# How Can Dupilumab Cause Eosinophilic Pneumonia?

**DOI:** 10.3390/biom12121743

**Published:** 2022-11-23

**Authors:** Momoko Kurihara, Katsunori Masaki, Emiko Matsuyama, Masato Fujioka, Reina Hayashi, Saki Tomiyasu, Kotaro Sasahara, Keeya Sunata, Masato Asaoka, Yuto Akiyama, Miyuki Nishie, Misato Irie, Takae Tanosaki, Hiroki Kabata, Koichi Fukunaga

**Affiliations:** 1Division of Pulmonary Medicine, Department of Medicine, School of Medicine, Keio University, Tokyo 160-8582, Japan; 2Keio Allergy Center, Keio University Hospital, Tokyo 160-8582, Japan; 3Department of Molecular Genetics, School of Medicine, Kitasato University, Kanagawa 252-0374, Japan; 4Clinical and Translational Research Center, Keio University Hospital, Tokyo 160-8582, Japan

**Keywords:** dupilumab, eosinophilic pneumonia, IL-4/13

## Abstract

Reports of eosinophilic pneumonia (EP) as a side effect of dupilumab administration are limited in previous studies. Herein, we report two cases in which EP developed subsequent to the administration of dupilumab for eosinophilic chronic rhinosinusitis (ECRS). Case 1: A 55-year-old woman presented with ECRS, eosinophilic otitis media, and bronchial asthma, and was treated with dupilumab for ECRS. Five weeks later, fever and dyspnea developed, and infiltration shadows were observed in her lungs. The peripheral blood eosinophil count (PBEC) was 3848/μL (26%), bronchoalveolar lavage fluid showed eosinophilic infiltration, and EP was subsequently diagnosed. Her condition improved following prednisolone treatment. Case 2: A 59-year-old man presented with fatigue and dyspnea after receiving dupilumab for ECRS. He had infiltrative shadows throughout his left lung field, and his PBEC was 4850/μL (26.5%). Prednisolone was initiated, and his condition improved. EP developed in both patients during the period of elevated PBEC after dupilumab administration, and dupilumab was suspected to be the causative agent in their EP. Hence, EP should be considered as a differential diagnosis when fever and dyspnea appear following dupilumab administration.

## 1. Introduction

Dupilumab is an IgG4 human monoclonal antibody that binds to IL-4Rα. Dupilumab inhibits interleukin (IL)-4 receptor (IL-4R) signaling induced by both IL-4 and IL-13. Dupilumab also controls inflammation in a variety of allergic disorders, including atopic dermatitis, asthma, eosinophilic chronic rhinosinusitis, and other allergic diseases [1,2]. Dupilumab inhibits the expression of vascular cell adhesion molecule-1 (VCAM-1) and the production of eotaxin via interleukin (IL)-4 and IL-13, thus inhibiting eosinophil migration into tissues, but its mechanism of action is opaque. Although eosinophil counts in peripheral blood may increase asymptomatically and transiently within the months following the introduction of dupilumab, few reports have shown that dupilumab causes eosinophilic pneumonia. In this study, we discuss eosinophilic pneumonia caused by dupilumab through the lens of two actual cases we have experienced.

## 2. Sharing Our Experience

First, we present two real cases that we experienced. A 55-year-old woman presented to our hospital with eosinophilic otitis media, eosinophilic chronic rhinosinusitis, and bronchial asthma. She complained of fever (37–38 °C) and dyspnea for one month. Seventeen days prior to admission, she had visited another hospital for possible community-acquired pneumonia and bronchitis, and had been treated with levofloxacin. There was no improvement in her symptoms. The patient was treated with inhaled indacaterol acetate/glycopyrronium bromide/high-dose mometasone furoate and montelukast for a prolonged period. Approximately 5 weeks before the onset of her symptoms, dupilumab was started for the treatment of eosinophilic chronic rhinosinusitis. She had no drug or food allergies and no lifestyle or environmental changes. She had no history of active or passive smoking. Prior to the introduction of dupilumab, she was not taking systemic steroids.

On arrival at our hospital, her blood pressure, pulse rate, and oxygen saturation were normal; however, her body temperature was 37.8 °C. Her respiratory rate was 16 breaths/min, and coarse crackles were heard bilaterally in her upper lung fields. Blood tests revealed leukocytosis (12,800/μL), eosinophilia (30.1%), and elevated C-reactive protein levels (7.39 mg/dL). The sialylated carbohydrate antigen KL-6 and surfactant protein D (SP-D) levels were normal. Anti-*Trichosporon asahii* antibody, interferon-gamma release assays, antineutrophil cytoplasmic antibody, angiotensin-converting enzyme, and anti-glycopeptidolipid core antibody were all negative. The polymerase chain reaction (PCR) test for coronavirus disease 2019 was negative. Because of prolonged symptoms, she was admitted to our hospital as an emergency case.

Her chest X-ray on admission showed consolidation in the bilateral peripheral-based upper lung fields and a typical radiographic pattern, i.e., “photographic negative of pulmonary edema” [3]. (Figure 1) Chest computed tomography (CT) revealed extensive ground-glass opacities in the left lung with contractile changes. Non-regional consolidation was scattered throughout the peripheral upper and middle lobes of the right lung (Figure 2).

On day 4 of hospitalization, bronchoscopy was performed, followed by bronchoalveolar lavage and a random transbronchial lung biopsy. Blood cell fractionation of the bronchoalveolar lavage fluid showed neutrophils (25.0%), eosinophils (25.5%), lymphocytes (32.5%), and monocytes (5.0%). Histopathological findings were consistent with the findings of organizing pneumonia with eosinophil infiltration. Treatment with prednisolone (50 mg/day, 1 mg/kg/day) was initiated on the fifth day of hospitalization. After the start of prednisolone treatment, her fever resolved and the shadows of the lung fields improved. The patient was discharged on day 13. The prednisolone dose was reduced to 40 mg/day at the time of discharge and to 30 mg/day one week later. Her pneumonia did not recur and progressed; thus, prednisolone was tapered in the outpatient clinic. When the prednisolone dose was reduced to 1 mg/day, six months after discharge, her nasal congestion worsened. Therefore, we considered the possibility of relapse of eosinophilic chronic rhinosinusitis and started benralizumab. After the initiation of benralizumab, her nasal congestion and asthmatic symptoms were stable.

The following is one more case. A 59-year-old man with eosinophilic chronic rhinosinusitis, without asthma, presented with dyspnea. After surgical excision of the nasal polyps, he inhaled fluticasone frankincarboxylic acid and took beclomethasone propionate nasally, but had not taken systemic steroids. Eleven weeks after the initiation of dupilumab for eosinophilic chronic rhinosinusitis at the otorhinolaryngology clinic, fever, fatigue, and dyspnea appeared and persisted. The patient was treated with ceftriaxone at the clinic; his symptoms did not improve. He was admitted to our hospital for close investigation and treatment 13 weeks after the onset of symptoms. His treatment consisted of only expectorants; no allergies were reported. He had no history of active or passive smoking.

On arrival at our hospital, his blood pressure, pulse rate, and oxygen saturation were normal; however, his body temperature was 37.7 °C. The patient’s respiratory rate was 18 breaths/min. Chest auscultation revealed coarse crackles throughout the left lung field. A chest X-ray showed consolidation in the left lung field. (Figure 3) A chest CT scan showed consolidation in the entire left lung field and leftward deviation of the mediastinum, due to contractile changes. A small area of consolidation was seen just below the peripheral pleura in the right upper and middle lobes. (Figure 4) His blood tests showed leukocytosis (18,300/μL), eosinophilia (26.5%), and high C-reactive protein levels (18.12 mg/dL). KL-6 and SP-D levels were normal. Tests for beta-D-glucan, aspergillus antigen, interferon-gamma release assays, antineutrophil cytoplasmic antibody, antinuclear antibody, anti-aminoacyl-tRNA synthetase antibody, Anti-Scl-70 antibody, and anti-cyclic citrullinated peptide antibody were all negative. The PCR test for coronavirus disease 2019 was negative.

Although the imaging findings were nonspecific, on the basis of the patient’s refractory course to antibacterial drugs and the high peripheral eosinophil counts, we considered this patient’s diagnosis to be consistent with eosinophilic pneumonia. After admission, prednisolone was started at 0.5 mg/kg/day, and was gradually decreased. The patient’s pneumonia findings and symptoms markedly improved, and the prednisolone dose was tapered. Unfortunately, one month after discontinuation of prednisolone, the eosinophilic pneumonia recurred, and he required oral prednisolone treatment. At relapse, the percentage of peripheral blood eosinophils had increased to 56%.

## 3. Mechanism of Eosinophil Increase with Dupilumab

We managed two cases of drug-induced eosinophilic pneumonia following dupilumab administration. Reports of EP after introduction of dupilumab is rare; however, a few have been reported in the past. Numata et al. reported that one of 26 Japanese patients with severe asthma who were treated with dupilumab developed hypereosinophilia, and required hospitalization for repetitive chronic eosinophilic pneumonia from April 2019 to December 2021 [4]. A previous phase III clinical trial, which evaluated the efficacy and safety of dupilumab, demonstrated that approximately 4% (52/1263) of adult patients with moderate-to-severe uncontrolled asthma had elevated blood eosinophil levels. Two of these patients developed severe eosinophilic pneumonia and required discontinuation of dupilumab [5].

It has been hypothesized that eosinophil counts in the peripheral blood increased after treatment with dupilumab [6,7]. Dupilumab inhibits the expression of vascular cell adhesion molecule-1 (VCAM-1) and the production of eotaxin via interleukin (IL)-4 and IL-13, thus inhibiting eosinophil migration into tissues. However, it does not prevent eosinophil migration from the bone marrow, resulting in an increase in peripheral blood eosinophil counts [8].

The patients with atopic dermatitis inducing dupilumab-associated conjunctivitis showed that the peripheral blood eosinophil count was markedly elevated [9]. This study suggests that organ damage involving eosinophils is associated with elevated peripheral eosinophil counts. In this report, we found that an increase in the peripheral blood eosinophilic count was associated with eosinophilic pneumonia development. However, the cause of peripheral blood eosinophil elevation and the trigger for organ-limiting eosinophil infiltration in the eyes and lungs remain unclear.

Nishiyama et al. reported that serum cytokine levels were measured in two cases of dupilumab-associated eosinophilic pneumonia, and elevated levels of CCL26 (eotaxin-3), periostin, and IL-5 [10] were found. In contrast, IL-5 was stable in patients who did not develop dupilumab-related eosinophilic pneumonia [5,6]. These findings suggest that the other activated IL-5-producing cells are associated with dupilumab-associated eosinophilic pneumonia. Further studies are needed to confirm the hypothesis that VCAM-1 is induced by pathways other than IL-4/13, and that there are other mechanisms of eosinophil adhesion besides VCAM-1 [10].

Compared to previous reports, the combination of the detailed clinical course and the fact that bronchoalveolar lavage test was performed are the novel findings. Nishiyama performed a bronchoscopic biopsy, but did not perform an bronchoalveolar lavage test. Numata had reported the similar case, but did not demonstrate the clinical course in detail.

In both cases, infection-related tests and autoantibodies, including ANCA, were negative at the time of diagnosis. Two patients did not have any organ damage, such as peripheral neuropathy, gastrointestinal tract damage, skin lesions, renal lesions, etc. The inflammation was localized to the lungs and airway, and, therefore, was unlikely to be eosinophilic granulomatosis with polyangiitis.

A survey by a pharmaceutical company on the side effects of dupilumab reported that eosinophilic pneumonia has been observed as a complication, and that attention should be paid to eosinophil counts and worsening pulmonary symptoms during treatment with dupilumab, especially during the first few months. 

A criterion has not been made regarding the continuation of dupilumab after the resolution of pneumonia. Additionally, reports have not been described regarding recurrence of adverse events after restarting dupilumab. Sudo et al. recommend that a combination of a low-dose steroids and dupilumab, after countering the side-effect of eosinophilic pneumonia induced by dupilumab alone, be used in order to continue injecting dupilumab [11]. However, since there is no established method of dealing with the disease, verification of pathological characteristics and factors that trigger the onset of EP, as well as verification of changes in peripheral blood eosinophils, are desirable in the future.

## 4. Conclusions

Eosinophilic pneumonia may occur during the period of elevated peripheral blood eosinophil counts after dupilumab administration. Dupilumab is suspected to be the cause of eosinophilic pneumonia. Therefore, eosinophilic pneumonia should be considered as a differential diagnosis in the presentation of fever or dyspnea after dupilumab administration. 

## Figures and Tables

**Figure 1 biomolecules-12-01743-f001:**
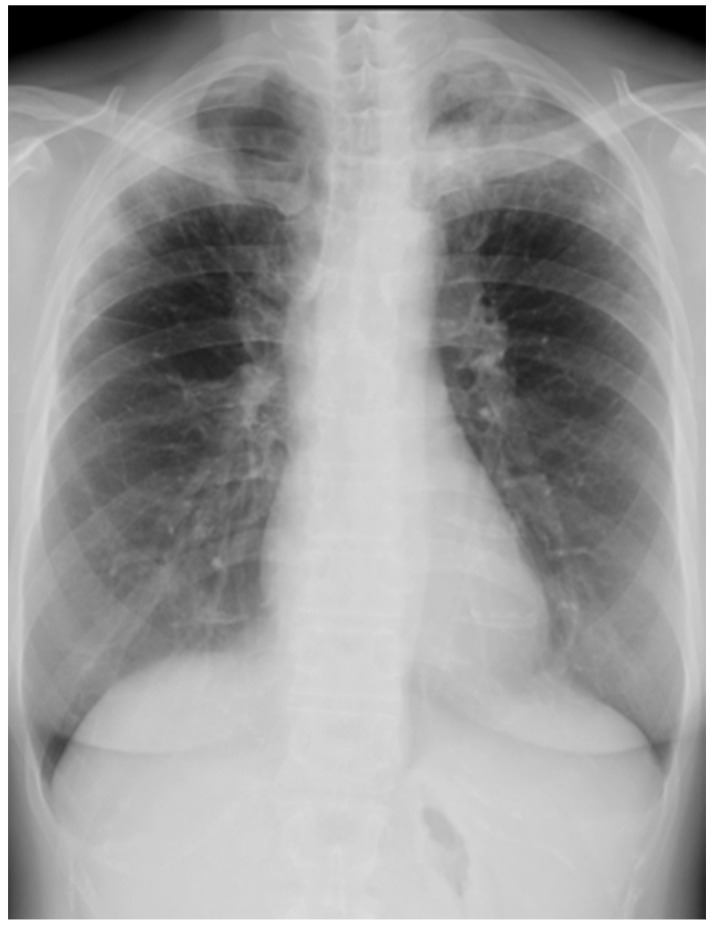
Chest X-ray of 55-year-old woman upon admission with fever and dyspnea. It showed bilateral consolidation in her upper lung fields.

**Figure 2 biomolecules-12-01743-f002:**
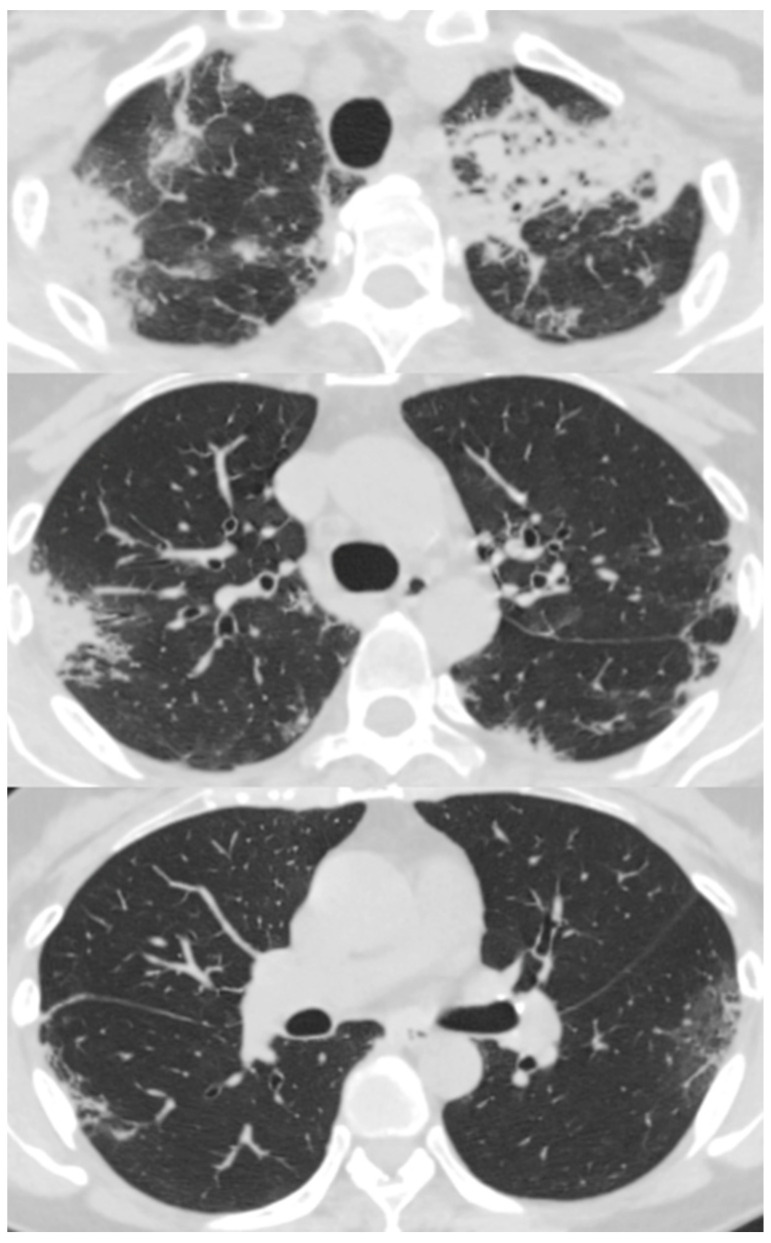
Chest computed tomography scan of 55-year-old woman with fever and dyspnea showed extensive ground-glass opacity in the left lung with contractile changes. Non-regional consolidation was scattered in the peripheral upper and middle lobes of her right lung.

**Figure 3 biomolecules-12-01743-f003:**
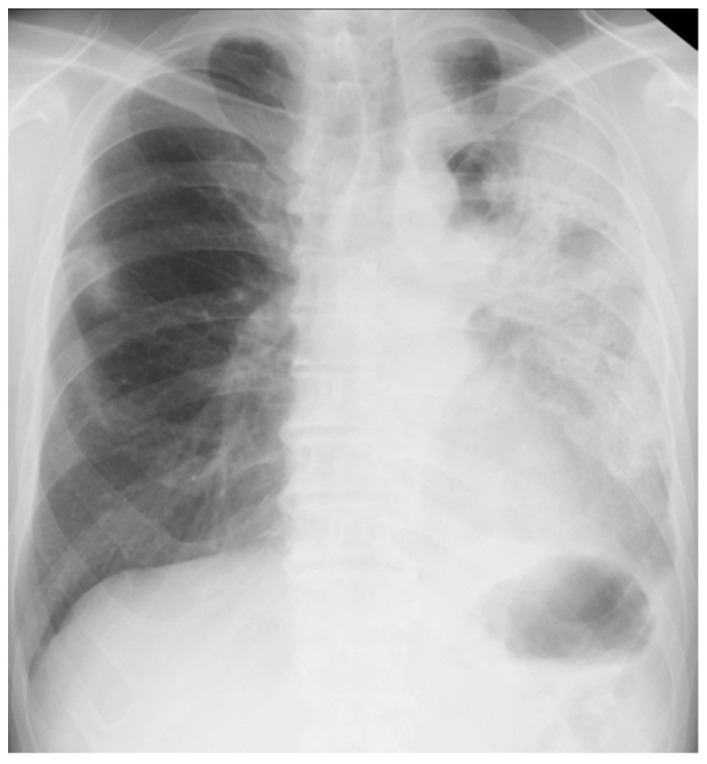
Chest X-ray of a 59-year-old man with dyspnea. It showed consolidation throughout his left lung field.

**Figure 4 biomolecules-12-01743-f004:**
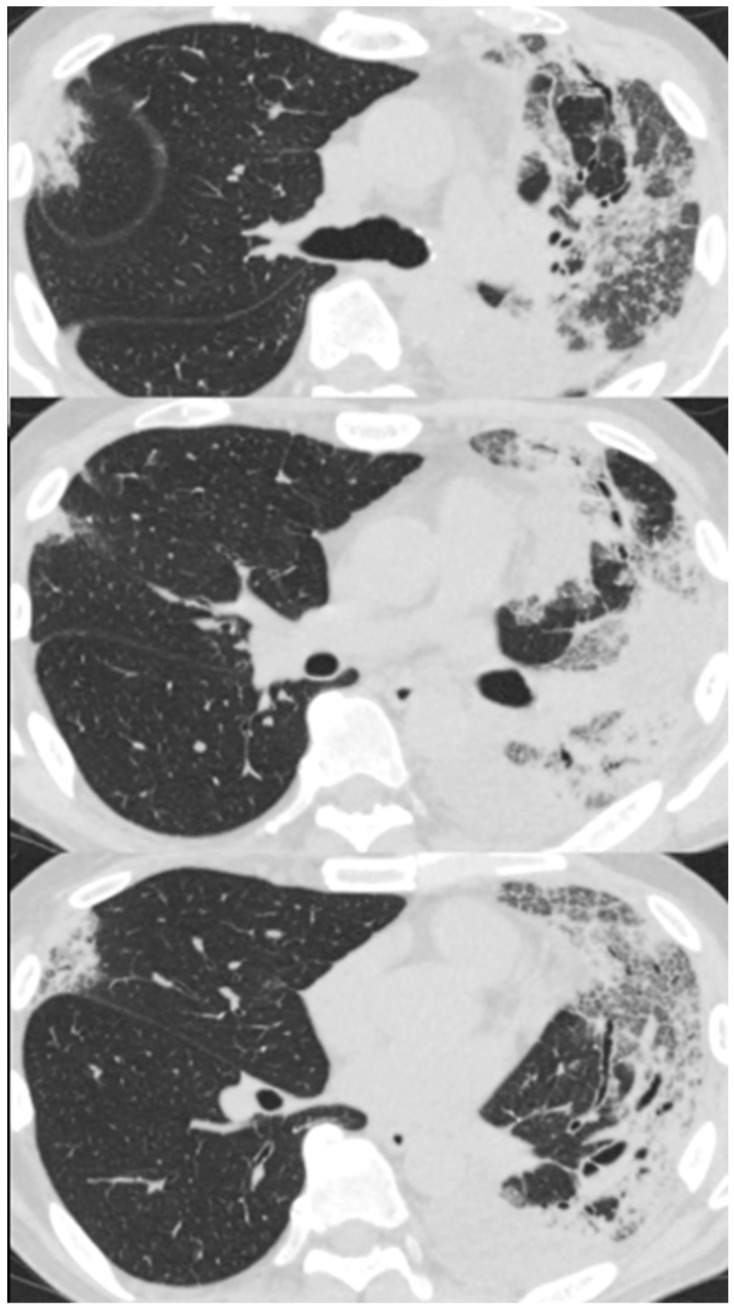
Chest computed tomography of a 59-year-old man with dyspnea scan showed consolidation in the entire left lung field and a leftward deviation of the mediastinum due to contractile changes. A slight consolidation was seen just below the peripheral pleura in the right upper and middle lobes.

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
