# Peer review of "How Can Dupilumab Cause Eosinophilic Pneumonia?"

_biomolecules, 2022, doi:10.3390/biom12121743_

Round 1

Reviewer 1 Report

See the attachment

Author Response

The eosinophilic pneumonia is well known, although rare, complication of dupilumab treatment; nevertheless, it is always interesting to find some new data in the field. The authors present two cases of eosinophilic pneumonia in patients treated with dupilumab and explain the mechanism of such adverse reaction.

I have some remarks regarding the manuscript:

RESPONSE:

Thank you for the reviewer’s comment.. We would like to thank the reviewer for his/her constructive critique to improve the manuscript. We have made every effort to address the issues raised and to respond to all the comments. The revisions are indicated highlighted in the revised hope that our revisions will meet the reviewer’s expectations.

  1. Some more data on dupilumab general mechanism of action should be moved to the “Introduction” from the third part of the manuscript

RESPONSE:

Thank you for the reviewer’s comment. I have revised the manuscript as written below;.

Dupilumab inhibits the expression of vascular cell adhesion molecule-1 (VCAM-1) and the production of eotaxin via interleukin (IL)-4 and IL-13, thus inhibiting eosinophil migration into tissues but its mechanism of action is opaque. (Lines 34-36)

  1. What was the treatment both patients received before the implementation of dupilumab? Did they receive systemic steroids? If yes, which dose and was the dose decreased after dupilumab was given?

RESPONSE:

In both cases, no oral steroids were taken prior to the start of Dupixent. I added the sentence in my manuscript.

Prior to the introduction of Dupixent, she was not taking systemic steroids. (Lines 52-53)

After surgical excision of the nasal polyps, he inhaled fluticasone frankincarboxylic acid and taking beclomethasone propionate nasally, but had not taken systemic steroids. (Lines 85-87)

  1. In the first patient, the effect of treatment with prednisolone is not apparent – “not worsened” also means “not improved” – a more thorough description and adding chest scans showing the resolution of changes would be reasonable.

RESPONSE:

We added the description “Her pneumonia did not recur and progressed; thus”. (Lines 78-79)

  1. The second patient provides no proof of the eosinophilic character of radiological changes. The presentation with dense consolidations and mediastinum shift is not typical for eosinophilic pneumonia. Again, we also do not have the results of prednisolone treatment showed (mitigation of symptoms only is mentioned)

RESPONSE:

We have added the explanation for this in the manuscript as written below:

Although the imaging findings were nonspecific, on the basis of the patient's refractory course to antibacterial drugs and the high peripheral eosinophil counts, we considered this patient's diagnosis consistent with eosinophilic pneumonia. After admission, prednisolone was started at 0.5 mg/kg/day and was gradually decreased. The patient's pneumonia findings and symptoms markedly improved, and the prednisolone dose was tapered. (Lines 107-112)

  1. Some revision of figures subtitles is needed – they are not clear enough

RESPONSE:

Thank you for the reviewer’s comment. We have added the explanation for this in the manuscript as written below:

Figure 1. Chest X-ray of 55-year-old woman on admission with fever and dyspnea. It showed bilateral consolidation in her upper lung fields. (Lines 117-118)

Figure 2. Chest computed tomography scan of 55-year-old woman with fever and dyspnea showed extensive ground-glass opacity in the left lung with contractile changes. Non-regional consolidation was scattered in the peripheral upper and middle lobes of her right lung.

Figure 3. Chest X-ray of 59-year-old man with dyspnea. It showed consolidation throughout his left lung field.

Figure 4. Chest computed tomography of 59-year-old man with dyspnea scan showed consolidation in the entire left lung field and a leftward deviation of the mediastinum due to contractile changes. A slight consolidation was seen just below the peripheral pleura in the right upper and middle lobes.

  1. Some improvement in language style across the case report should be considered

RESPONSE:

With the advice of an English editing specialist, we improved language style of the manuscript.

Reviewer 2 Report

1.     General comments

The authors described two cases of eosinophilic pneumonia during the course of dupilumab treatment for eosinophilic chronic rhinosinusitis (ECRS).

In the first case, 5 weeks after starting dupilumab treatment, the patient presented with fever, increased peripheral blood eosinophils and elevated CRP, as well as ground-glass opacities in bilateral lungs on CT, and bronchoscopy was diagnosed as organizing pneumonia with increased eosinophils in alveolar lavage fluid. The patient improved with steroid therapy, but the ECRS flared up during the steroid reduction process, and the patient was treated with benralizumab.

In the second case, fever, general fatigue, and dyspnea appeared 11 weeks after the start of dupilumab treatment, and 13 weeks after the onset, the patient was diagnosed with chronic eosinophilic pneumonia (CEP) based on peripheral blood eosinophilia, elevated CRP, and CT findings. CEP improved with steroid therapy, but after the steroid was discontinued, CEP flared up again, requiring steroid therapy again.

  The authors discuss the mechanism of eosinophilia that occurs in tissues during treatment with dupilumab in these cases, noting that there may be a mechanism for increased CCL26 and IL-5 production and VCAM-1 expression without IL-4/13 signaling from previous reports.

 Although this report is considered a case series that contributes to the development of clinical allergology in that it showed that eosinophilic pneumonia can occur during dupilumab treatment, it requires revision in several respects.

2.     Specific comments

a)     Major:

1)    Both cases have in common that they developed eosinophilic pneumonia during dupilumab treatment and relapsed after steroid reduction, which is an interesting point in the disease course. However, it is necessary to clearly describe what is novel in these two cases compared to the reports of Numata et al. and Nishiyama et al.

2)    In both cases, although infection-related tests and autoantibodies including ANCA were negative at the time of diagnosis, there is insufficient differentiation between eosinophilic pneumonia and eosinophilic granulomatosis with polyangiitis (EGPA) EGPA. It would be better to specify the presence or absence of organ damage such as peripheral neuropathy, gastrointestinal tract damage, skin lesions, renal lesions, etc., and to clarify that it is eosinophilic pneumonia and not EGPA.

b)    Minor:

1)    eosinophilic chronic rhinosinusitis, eosinophilic chronic sinusitis, eosinophilic chronic sinusitis with nasal polyps, eosinophilic sinusitis, etc. The terminology for eosinophilic chronic rhinosinusitis is not standardized. It is recommended that the notation be standardized to either eosinophilic chronic rhinosinusitis or chronic sinusitis with eosinophilic nasal polyps.

2)    Line 132; Since eotaxin is a chemokine, "production" is sufficient, but since VCAM-1 is a cell surface adhesion molecule, "expression" should be used.

3)    Line 137-140; The sentence is difficult to understand. Are the authors trying to describe that peripheral blood eosinophil counts were increased in both groups with and without dupilumab-associated conjunctivitis?

4)    Line 140-143; The same sentences are repeated and are redundant.

5)    Line 145-151; The sentence is difficult to understand.

The authors describe the results of Nishiyama et al.'s report as consistent with previous reports. However, Nishiyama et al. reported elevated levels of IL-5, CCL26, and periostin, whereas Castro et al. reported decreased levels of CCL26 and periostin in patients treated with dupilumab (ref. 5). It is recommended that the sentence be corrected in this regard.

Do the authors wish to convey the following in this sentence?

“Nishiyama et al. reported that serum cytokine levels were measured in two cases of dupilumab-associated eosinophilic pneumonia and found elevated levels of CCL26 (eotaxin-3), periostin, and IL-5. In contrast, IL-5 was stable in patients who did not develop dupilumab-related eosinophilic pneumonia. Those findings suggest that the other activated IL-5-producing cells are associated with dupilumab-associated eosinophilic pneumonia. “

6)    If abbreviations are used, the full name should be provided at the time of first appearance.

Line 30; IL-4R → interleukin (IL)-4 receptor (IL-4R)

Line 147; CCL26 → C-C motif chemokine ligand 26 (CCL26)

7)    Spelling errors and grammatical expressions need correction.

Line 95; -D-glucan → beta-D-glucan or β-D-glucan

Line 123; Numata et al. report that ~ → Numata et al. reported that ~

Line 173-171; Dupilumab is suspected to be the cause eosinophilic pneumonia.→Dupilumab is suspected to be the cause of eosinophilic pneumonia.

Author Response

  1. General comments

The authors described two cases of eosinophilic pneumonia during the course of dupilumab treatment for eosinophilic chronic rhinosinusitis (ECRS).

In the first case, 5 weeks after starting dupilumab treatment, the patient presented with fever, increased peripheral blood eosinophils and elevated CRP, as well as ground-glass opacities in bilateral lungs on CT, and bronchoscopy was diagnosed as organizing pneumonia with increased eosinophils in alveolar lavage fluid. The patient improved with steroid therapy, but the ECRS flared up during the steroid reduction process, and the patient was treated with benralizumab.

In the second case, fever, general fatigue, and dyspnea appeared 11 weeks after the start of dupilumab treatment, and 13 weeks after the onset, the patient was diagnosed with chronic eosinophilic pneumonia (CEP) based on peripheral blood eosinophilia, elevated CRP, and CT findings. CEP improved with steroid therapy, but after the steroid was discontinued, CEP flared up again, requiring steroid therapy again.

  The authors discuss the mechanism of eosinophilia that occurs in tissues during treatment with dupilumab in these cases, noting that there may be a mechanism for increased CCL26 and IL-5 production and VCAM-1 expression without IL-4/13 signaling from previous reports.

 Although this report is considered a case series that contributes to the development of clinical allergology in that it showed that eosinophilic pneumonia can occur during dupilumab treatment, it requires revision in several respects.

RESPONSE:

We would like to thank the reviewers for their constructive critique to improve the manuscript. We have made every effort to address the issues raised and to respond to all the comments. The revisions are indicated below and highlighted in the revised manuscript. We hope that our revisions will meet the reviewers’ expectations.

  1. Specific comments
  2. a) Major:

1)    Both cases have in common that they developed eosinophilic pneumonia during dupilumab treatment and relapsed after steroid reduction, which is an interesting point in the disease course. However, it is necessary to clearly describe what is novel in these two cases compared to the reports of Numata et al. and Nishiyama et al.

RESPONSE:

We have added the explanation for this in the manuscript as written below:

“Compared to previous reports, the combination of the detailed clinical course and the fact that bronchoalveolar lavage test was performed are the novel finding. Nishiyama performed a bronchoscopic biopsy but did not perform an bronchoalveolar lavage test.  Numata had reported the similar case but did not demonstrate clinical course in detail.” (Lines 160-163)

2)    In both cases, although infection-related tests and autoantibodies including ANCA were negative at the time of diagnosis, there is insufficient differentiation between eosinophilic pneumonia and eosinophilic granulomatosis with polyangiitis (EGPA) EGPA. It would be better to specify the presence or absence of organ damage such as peripheral neuropathy, gastrointestinal tract damage, skin lesions, renal lesions, etc., and to clarify that it is eosinophilic pneumonia and not EGPA.

RESPONSE:

Thank you for the reviewer’s constructive comment. We have added the explanation for this in the manuscript as written below:

“In both cases, infection-related tests and autoantibodies including ANCA were negative at the time of diagnosis. two patients did not have any organ damage such as peripheral neuropathy, gastrointestinal tract damage, skin lesions, renal lesions, etc. The inflammation was localized to the lungs and airway, therefore, unlikely to be eosinophilic granulomatosis with polyangiitis.” (Line 164-168)

  1. b) Minor:

1)    eosinophilic chronic rhinosinusitis, eosinophilic chronic sinusitis, eosinophilic chronic sinusitis with nasal polyps, eosinophilic sinusitis, etc. The terminology for eosinophilic chronic rhinosinusitis is not standardized. It is recommended that the notation be standardized to either eosinophilic chronic rhinosinusitis or chronic sinusitis with eosinophilic nasal polyps.

RESPONSE:

We have unified these terminologies on “eosinophilic chronic rhinosinusitis”.

2)    Line 132; Since eotaxin is a chemokine, "production" is sufficient, but since VCAM-1 is a cell surface adhesion molecule, "expression" should be used.

RESPONSE:

Thank you for the reviewer’s comment. We have added the explanation for this in the manuscript as written below:

Dupilumab inhibits the expression of vascular cell adhesion molecule-1 (VCAM-1) and the production of eotaxin via interleukin (IL)-4 and IL-13, thus inhibiting eosinophil migration into tissues. (Line 140-142)

3)    Line 137-140; The sentence is difficult to understand. Are the authors trying to describe that peripheral blood eosinophil counts were increased in both groups with and without dupilumab-associated conjunctivitis?

RESPONSE:

We have added the explanation for this in the manuscript as written below:

The patients with atopic dermatitis inducing dupilumab-associated conjunctivitis showed that the peripheral blood eosinophil count was markedly elevated. This study suggests that organ damage involving eosinophils is associated with elevated peripheral eosinophil counts. (Line 145-148)

4)    Line 140-143; The same sentences are repeated and are redundant.

RESPONSE:

Thank you for the reviewer’s comment. We have removed some sentences and improved the writing style according to the English editing specialist.

5)    Line 145-151; The sentence is difficult to understand.

The authors describe the results of Nishiyama et al.'s report as consistent with previous reports. However, Nishiyama et al. reported elevated levels of IL-5, CCL26, and periostin, whereas Castro et al. reported decreased levels of CCL26 and periostin in patients treated with dupilumab (ref. 5). It is recommended that the sentence be corrected in this regard.

Do the authors wish to convey the following in this sentence?

“Nishiyama et al. reported that serum cytokine levels were measured in two cases of dupilumab-associated eosinophilic pneumonia and found elevated levels of CCL26 (eotaxin-3), periostin, and IL-5. In contrast, IL-5 was stable in patients who did not develop dupilumab-related eosinophilic pneumonia. Those findings suggest that the other activated IL-5-producing cells are associated with dupilumab-associated eosinophilic pneumonia. “

RESPONSE:

Thank you for the reviewer’s constructive comment. We have added the explanation for this in the manuscript as written below:

Nishiyama et al. reported that serum cytokine levels were measured in two cases of dupilumab-associated eosinophilic pneumonia and found elevated levels of CCL26 (eotaxin-3), periostin, and IL-5 [10]. In contrast, IL-5 was stable in patients who did not develop dupilumab-related eosinophilic pneumonia [5, 6]. Those findings suggest that the other activated IL-5-producing cells are associated with dupilumab-associated eosinophilic pneumonia. Further studies are needed to confirm the hypothesis that VCAM-1 is induced by pathways other than IL-4/13, and that there are other mechanisms of eosinophil adhesion besides VCAM-1 [10]. (Line 152-159)

6)    If abbreviations are used, the full name should be provided at the time of first appearance.

Line 30; IL-4R → interleukin (IL)-4 receptor (IL-4R)

Line 147; CCL26 → C-C motif chemokine ligand 26 (CCL26)

RESPONSE:

Thank you for the reviewer’s comment. We have fixed them.

7)    Spelling errors and grammatical expressions need correction.

Line 95; -D-glucan → beta-D-glucan or β-D-glucan

Line 123; Numata et al. report that ~ → Numata et al. reported that ~

Line 173-171; Dupilumab is suspected to be the cause eosinophilic pneumonia.→Dupilumab is suspected to be the cause of eosinophilic pneumonia.

RESPONSE:

Thank you for the reviewer’s comment. We have fixed them.

Round 2

Reviewer 2 Report

 I have confirmed that the authors have responded appropriately to the points raised. I recommend that the manuscript be accepted with the following minor corrections.

Line 52-53; eosinophilic chronic rhinosinusitis with nasal polyps

→ eosinophilic chronic rhinosinusitis

Line 54-55; Prior to the introduction of Dupixent,

→ Prior to the introduction of dupilumab,

Author Response

I have confirmed that the authors have responded appropriately to the points raised. I recommend that the manuscript be accepted with the following minor corrections.

RESPONSE:

We would like to thank the reviewer for his/her comments to improve the manuscript.

Line 52-53; eosinophilic chronic rhinosinusitis with nasal polyps

→ eosinophilic chronic rhinosinusitis

RESPONSE:

Thank you. We fixed the manuscript as written below:

“Approximately 5 weeks before the onset of her symptoms, dupilumab was started for the treatment of eosinophilic chronic rhinosinusitis.” (Lines 49-50)

Line 54-55; Prior to the introduction of Dupixent,

→ Prior to the introduction of dupilumab,

RESPONSE:

Thank you. We fixed the manuscript as written below:

“Prior to the introduction of dupilumab, she was not taking systemic steroids.” (Lines 52-53)
